# Computational analysis of 3D biopolymeric porous scaffolds for the *in vitro* development of neural networks

Ilaria Parodi[1,2*], Laura Pastorino[1], Giacomo Damonte[3], Silvia Scaglione[2,4], Marco Massimo Fato[1,2], Donatella Di Lisa[1]

1 Department of Informatics, Bioengineering, Robotics and System Engineering, University of Genoa, Genoa, Italy, 2 National Research Council of Italy, Institute of Electronic, Computer and Telecommunications Engineering (IEIIT-CNR), Genoa, Italy, 3 Department of Chemistry and Industrial Chemistry, University of Genoa, Genoa, Italy, 4 React4life S.p.A., Genoa, Italy

* ilaria.parodi@edu.unige.it, ila.parodi96@gmail.com

## Abstract

Computational modelling can be used to study and improve specific experimental tissue engineering protocols and outcomes. Proper oxygenation and nutrient substances supply such as glucose are crucial in 3D *in vitro* models. In most cases, hydrogel-based scaffolds are employed as culture systems. However, the diffusion of molecules could be difficult in the innermost areas of the scaffolds, and the presence of gradients could affect cell proliferation, especially in static conditions. Hence, the mathematical modelling of oxygen and nutrient transport, as well as their consumption by the expanding cell culture within the scaffold, can be useful for optimizing tissue construct properties and generating more predictive and robust outcomes. In this work, nutrient diffusion has been studied through two different scaffolds seeded with glial and neuronal cells: chitosan microbeads and PLA fibers covered in chitosan produced with two specific fabrication-based techniques. First, homogenization theory has been applied to the two different porous constructs, formulated as heterogeneous domains composed of two distinct phases: the culture medium and the polymeric material. Then, a continuous mathematical model of nutrient transportation-consumption and cell proliferation has been implemented in COMSOL, aiming to comprehend nutrient diffusion to allow suitable environmental conditions for the growth of neural cells on 3D biopolymeric scaffolds.

## Introduction

Three-dimensional (3D) *in vitro* models are fundamental tools in tissue engineering, disease modelling and drug testing. Differently from two-dimensional (2D) systems, where cells grow in an unnatural spatial environment, characterized by a uniform exposure to culture medium and nutrient supply, in 3D cultures, cells have different

**Data availability statement:** All relevant data are within the manuscript.

**Funding:** This work was supported by the NextGenerationEU (NGEU) National Recovery and Resilience Plan (NRRP) within the Robotics and AI for Socio-economic Empowerment (RAISE) ecosystem – SPOKE 2.

**Competing interests:** The authors have declared that no competing interests exist.

access to nutrients, in a way strictly dependent on the size and the microarchitecture of the *in vitro* model [1]. In the human body, the vasculature plays a crucial role in supplying cells with adequate oxygen and nutrients while also facilitating the removal of metabolic waste: specifically, cells close to the blood vessels are exposed to high levels of oxygen, glucose and many other nutrients, while those further away perceive lower levels. Traditionally, the absence of an effective vascular network has been a critical barrier to scaling up three-dimensional *in vitro* tissue-engineered structures. For example, scaffold-free models such as spheroids and organoids represent self-assembling cell aggregates that grow in an environment that prevents attachment to a flat surface, which also guarantees intensive cell-cell contacts [2,3]. However, it has been shown that avascular organoids and spheroids with a size of 100 μm or larger are characterized by poor cell viability and functionality.

Another configuration of 3D culture is represented by scaffolds-based models, which can be produced by different techniques [4–14]. These systems provide which represent suitable microenvironments that support the cell growth in a native-like fashion [15–17], mimicking the extracellular matrix of specific tissues and allowing the diffusion of nutrients.

In this context, the scaffold porosity is crucial for ensuring adequate oxygenation and proper transport of nutrients [18]. Maintaining physiological oxygen levels is particularly important when working with sensitive cell types, as hypoxic conditions can compromise viability. However, in models aiming to replicate pathophysiological conditions (e.g., cancer, stroke, or Parkinson's disease), a more severe and restrictive environment may be desirable [19].

The problem of hypoxia can be addressed by using perfusion systems (such as bioreactors or microfluidic systems), to promote efficient nutrient exchange, waste removal, as well as the application of appropriate and controllable physicochemical stimuli, ensuring a continuous and spatially uniform supply of nutrients and gases to the cells throughout the scaffold domain. Differently, in static cultures, the diffusion into the innermost parts of the scaffold can be challenging due to the large size of the structure. Moreover, accurate measurements of the oxygen and glucose levels, particularly important to maintain long-term cell viability, could be invasive and interfere with the culture system, therefore not always applicable.

Similarly assessing cell proliferation, particularly in 3D constructs, is far from straightforward. In this case, computational analysis not only provides valuable information on the distribution of nutrients but also offers essential insights for designing a 3D model with characteristics and specifications that more accurately align with a truly biomimetic approach, in terms of porosity and thickness. In developing mathematical models for cell cultures, two primary approaches are commonly used [20]. The discrete approach, also known as the cell-based model, describes the interactions of each cell within the system by monitoring their specific positions and velocities. However, due to the complexity of tracking each cell independently, these models require sophisticated numerical methods for their solution [20,21]. Alternatively, a continuous approach considers the cell system as a unified domain where cell positions and proliferation rates are averaged and spatially distributed.

Represented through systems of coupled partial differential equations (PDEs), this approach is computationally more efficient, though it is less precise compared to cell-based models.

Different laws have been developed to describe cell proliferation, following a continuum approach: the exponential law well represents the simplest model for describing cell growth, with its growth rate also defined as "fitness", although it is valid only during the initial phase [22]. This kind of model, employed as an example by Gonzales *et al.*, is suitable for well-controlled single-species experiments, where daily medium refreshments are applied to maintain the cell culture consistently proliferating. However it does not consider the saturation of the culture space (petri-dish for 2D cultures and scaffold and other 3D systems) [23]; in Monod model, the growth rate depends on a limiting resource following the Michaelis–Menten (MM) kinetics; Maldonado *et al*. use it to estimate the growth of a bacteria in a bioreactor, while Coletti *et al*. developed a mathematical model for rat myoblasts in collagen gel through the Contois equation, a modified version of MM that take into accounts the contact inhibition [24,25]; the logistic law incorporates the carrying capacity-limited proliferation (*K*), successfully employed into the investigation of *in vivo* tumour progression, such as in the work of Penisson *et al.*, where a mathematical model was developed to predict the number of endogenous somatic mutations present in various tissue types of a patient at a given age, or in the study of Namdev *et al.* about the dynamics of tumour growth in the lung [26–28]; it has been used also for describing healthy tissue growth, like in the work of Shakeel, that modelled the cell proliferation in a perfused scaffold, in the analysis of El-Hackem *et al.* where the cell growth was put in relation with an ECM deposition, or in the model of Buenzli *et al*., developed for osteoblastic cells seeded on polycaprolactone (PCL) scaffolds [29,30], but also for wound healing: in this regard Trejo and Kujouharov investigated how phagocytes and inflammatory cytokines influence the early stages of *in vivo* bone fracture healing [31].

Another important aspect in the mathematical modelling of engineered tissues is to recapitulate the nutrient transport within the constructs. Different formulations have been used to model nutrient consumption by cell activity. In some cases, it is modelled as a constant value [18]. Androjna *et al.* investigated the oxygen transport through natural ECM in a bioreactor, assuming a zero-order kinetics for the gas consumption by the cells. Other models follow first-order kinetics both for the nutrient concentration and for the cell density, as in the study of Shakeel or in the work of Jeong *et al.*, analysing the growth of preosteoblastic cells in poly (lactic-co-glycolic acid) (PLGA) multi-layer scaffolds [29,32]. In others, it is represented as dependent on oxygen or glucose concentration, typically following the MM kinetics, a model shown to be effective across various tissues, both healthy and cancerous. In general, the sigmoidal form of MM kinetics aims to capture the lower OCR in a certain tissue under hypoxic conditions [33] than under normoxia. To this purpose, Grimes *et al.* investigated the oxygen depletion in tumour tissues moving away from blood vessels. Differently, Mofrad *et al.* used this kinetics to predict the oxygenation in a channelled perfused scaffold for myocardium. In another study, Shariati *et al.* observed the diffusivity-driven mass transfer of oxygen, glucose and insulin in a hydrogel laden with islet cells for the treatment of diabetes. Furthermore, Carrol *et al.* simulated the oxygen level in an agarose gel seeded with stem cells and chondrocytes [34–37].

However, the complex microstructure of 3D constructs can significantly increase the computational load, making simulations particularly time-consuming. Moreover, materials composed of two or more constituents, known as composite materials, exhibit heterogeneity at the microscale and require treatment as "effective" continua for analysis on the macroscale. In this sense, homogenization procedures based on microscopic considerations can be employed to derive constitutive relationships describing the properties of and interactions between the two material phases. Homogenization theory is typically applied in material science, especially in composite materials, to obtain macroscopical effective mechanical properties. Based on the underlying formulation, the homogenization methods can be divided into two main classes: analytical methods and numerical methods. Analytical methods offer significant advantages in terms of speed. However, they may lack accuracy when dealing with complex geometries or non-linear behaviours, as they rely on simplifications that do not always capture real-world complexity. Differently, numerical methods offer high accuracy, providing detailed, high-fidelity results that are particularly effective for

composites with complex microstructures. Moreover, they can be applied to a wide range of materials and structures, accommodating non-linear and heterogeneous composites that are difficult to handle analytically. However, these benefits come with certain drawbacks. The computational cost can be significant, as simulations are often time-consuming and resource-intensive, especially when dealing with 3D models or fine meshes [38,39]. The most common numerical approach involves Representative Volume Elements (RVE), where a small sample volume that statistically represents the material's microstructure is modelled and boundary conditions are applied, and properties are computed using simulations such as finite element analysis (FEA). In this context, Mei and Vernescu developed an RVE-based method aiming to solve the heat transport in composite materials [40]. To the best of our knowledge, the application of homogenization theory is quite limited in modelling tissue-engineered constructs. In particular, there are few papers related to the calculation of mechanical properties of the scaffold, such as the effective Young modulus [41,42], As regards the diffusive phenomena, Kojic *et al.* applied the homogenization theory to a heterogeneous structure representing the capillary bed first in 2D [43] and then in 3D [44].

This study applies homogenization theory to evaluate nutrient diffusion—specifically oxygen and glucose—in porous scaffolds tailored for neural tissue engineering, a highly sensitive and metabolically demanding environment. We implemented the homogenization method using COMSOL Multiphysics®, enabling the calculation of effective diffusivity tensors in two distinct scaffold configurations with complex microarchitectures. This mathematical framework significantly reduces computational costs by avoiding high-resolution microscale simulations while preserving accuracy in capturing the macroscopic behaviour of nutrient transport. The two scaffold types considered in this study were fabricated using a bottom-up assembly approach: one based on chitosan microbeads, previously developed by Tedesco *et al.* [45], and the other comprising chitosan-coated PLA fibrous layers, introduced in the present work. Both exhibit intricate and highly heterogeneous microstructures, which make them ideal candidates for the application of homogenization theory. Then we investigated oxygen and glucose transport in the two different 3D homogenized constructs as a function of cell density through a combined experimental-computational approach, aiming to maintain specific oxygen levels. In particular, the model presented in this study was applied to primary neuronal cells due to their greater sensitivity compared to cell lines or other tissues, ensuring suitable environmental conditions for the development of glial-neuronal cell culture.

## Materials and methods

### Cell culture

The cells were kindly provided in frozen vials. They were obtained from cortices, dissected, and extracted from embryonic Sprague–Dawley rats on gestational day 18, following sterile procedures. The experimental protocol was conducted in accordance with the guidelines set forth by the European Animal Care Legislation (2010/63/EU), as well as the approval of the Italian Ministry of Health, following the provisions of DL 116/1992. Additionally, the research adhered to the guidelines of the University of Genova, with a focus on reducing the number of animals used for the project and minimizing any potential suffering.

The fetal tissue was then dissociated into single cells through enzymatic digestion using 0.125% trypsin in Hank's balanced salt solution (Gibco Invitrogen) free of $Ca^{2+}$ and $Mg^{2+}$ for 20 minutes at 37°C. The enzymatic process was stopped by adding a culture medium supplemented with 10% FBS.

Subsequently, the cortical tissue was mechanically dissociated using a Pasteur pipette. The neural cells, consisting of neurons and glial cells, were suspended in a culture medium composed of NEUROBASAL (Gibco Invitrogen) supplemented with 2% (w/v) B-27 Supplement (Gibco Invitrogen), 1% Glutamax (Gibco Invitrogen), and 1% Pen-Strep (Gibco Invitrogen). The neuronal cultures, seeded on PLA fibers, were maintained in an incubator at 37°C with 5% $CO_2$ and 95% R.H. for three weeks, with half of the medium replaced once a week. The PLA films seeded with glial and neuronal cells were then fixed after one day and at the end of the culture.

### *In vitro* 3D scaffold-based models for neuronal culture

The first type of scaffold considered in this study was an assembly of chitosan microbeads, obtained in a previous study carried out by Tedesco *et al.* through the aerodynamically assisted bio-jetting [45]. The microbeads, mostly of which were characterized by a size of approximately 80 μm (radius ~ 40 μm), were poured in a 650 μm thick PDMS ring to realize a scaffold with a specific volume for a standard culture. Moreover, the constraint helps maintain the shape of the in vitro construct, which is further stabilized by the cells growing between the microbeads, contributing to their cohesion. For the 3D cell culture based on the chitosan microbeads, a first study has been conducted considering $3 \times 10^4$ microbeads and $1.2 \times 10^5$ total cells.

The other type of scaffold was a stacking of fibrous PLA films, fabricated using the electrospinning technique, coated with chitosan. Poly (L-lactic acid) (PLLA) and poly (D-lactic acid) (PDLA) (Synterra 1010, M.w. ~ 100 kDa) were dissolved at room temperature in a solvent mixture of chloroform ($CHCl_3$) and hexafluoro-2-propanol (HFIP) (1:2 ratio) at an 8% w/w concentration. Electrospinning was performed using a Gamma High Voltage Research Power Supply (model ES30P-5W), a Harvard Apparatus Model 44 Programmable Syringe Pump with a flow rate of 0.5 ml/h, and a grounded aluminium collector set at 20 cm of distance. in a Teflon chamber with humidity control. To promote the adhesion of nerve cells to the electrospun substrate, a 0.5% of chitosan solution in 0.1 M acetic acid was forced through the PLA film using a vacuum pump, with 20 ml perfused on each side. Excess chitosan was removed by perfusing the films with 20 ml of water, followed by an overnight drying process at 40°C. Finally, cell-supportive 60 μm-thick PLA films were obtained.

To evaluate the actual coverage of PLA electrospun fibers with chitosan, infrared absorption analyses were conducted on the films and compared with the singular chitosan and PLA. The spectra were acquired using an IR spectrometer (Bruker's FT-IR VERTEX 70v), operating between 400 and 4000 cm$^{-1}$, performing 128 scans (resolution: 2).

Morphological characterization of PLA fibers was performed using scanning electron microscopy (SEM) with a Hitachi S-2500 instrument (Hitachi, Chiyoda, Tokyo, Japan) operating in secondary electron mode at 10 kV. Samples were examined at different magnifications after being coated with a ~ 30 nm gold film using a Polaron sputter coater (Thermo VG Scientific, East Grinstead, UK). For the *in vitro* cell culture, the PLA film was cut into a circular section to fit in the PDMS ring and, subsequently, seeded with $3 \times 10^4$ cells.

### Homogenization theory

Diffusive transport of nutrients within the heterogeneous scaffolds takes place both inside the polymeric material and in the culture medium that surrounds the scaffolds. To conduct simpler computational analysis, the homogenization theory presented by Mei and Vernescu [40] has been implemented in COMSOL Multiphysics 6.0. The aim was to solve a problem at the micro-scale to obtain an effective diffusion tensor at the macroscale. At the micro-scale, it is possible to identify a representative volume element (RVE) $\Omega$, which is periodically repeated, composed of two different materials, where $\Omega_1$ and $\Omega_2$ are respectively the volumes occupied by the polymeric material (chitosan microbeads or PLA fibers) and the culture medium, characterized by two different diffusivities $D_1$ and $D_2$. By solving a set of partial differential equations (PDEs) on the representative volume element (RVE), using multiple-scale asymptotic expansions and imposing periodic boundary conditions, it is possible to find the components of the diffusivity tensor $\boldsymbol{D}$, which describe the diffusion properties of the homogenized material. Each component can be written as follows:

$$D_{ij} = \frac{1}{\Omega} \left[ \iiint_{\Omega_1} D_1 \left( \delta_{jk} + \frac{\partial w_{1_k}}{\partial x_j} \right) d\Omega + \iiint_{\Omega_2} D_2 \left( \delta_{jk} + \frac{\partial w_{2_k}}{\partial x_j} \right) d\Omega \right]$$

(1)

where the coefficients $(w_a)_i$ are the components of an unknown vector $\boldsymbol{w}$ periodic on $\Omega$ which must satisfy the following boundary conditions at the interface between the two materials:

$$w_{1_i} = w_{2_i}; \qquad D_1 \left( \delta_{ij} + \frac{\partial w_{1_j}}{\partial x_i} \right) n_i = D_2 \left( \delta_{ij} + \frac{\partial w_{2_j}}{\partial x_i} \right) n_i$$

(2)

The equations and the respective boundary and periodic conditions have been written in the *Weak Form PDE* interface, to calculate **w1** and **w2** and subsequently obtain the components $D_{ij}$ of the diffusion tensor (Eq (1)). A fine tetrahedral mesh was generated. In the following subsections the unit cells considered for the two kinds of polymeric scaffolds are described with their specific geometry and conditions.

**RVE identification and conditions.** To apply the homogenization theory to the microbeads scaffold, two different configurations were considered: the *body-centred cubic* (*bcc*) and the *face-centred cubic* (*fcc*) allowing to obtain a maximum packing density $\eta$ of 68% and 74% respectively (Fig 1A) The radius of the spheres has been estimated from the following expression

$$\frac{4}{3}\pi R^3 n_{beads} = \eta V_{tot}$$

(3)

where the number of microbeads $n_{beads}$ was set according to the experimental set up and $V_{tot}$ is the PDMS mold volume. The length *l* of the side of corresponding cubic RVEs has been obtained as follows:

$$l = \frac{4}{\sqrt{3}} \quad (bcc), \qquad l = \sqrt{2}R \quad (fcc)$$

(4)

To apply the homogenization theory to a scaffold of electrospun PLA fibers, despite the randomic deposition produced by this technique, a parallelepiped containing two equal cylinders overlapping at a right angle was chosen as the unit cell (Fig 1C). The height of the parallelepiped *h* has been set to be four times the radius *R* of the cylinders, as the fibers tend to be distributed on overlapping parallel planes; the length of the side of the parallelepiped *l* has been obtained as follows, considering the porosity $\phi$ of the scaffold

$$l = \frac{2\pi R^2}{(1-\phi)h} = \frac{\pi R}{2(1-\phi)}$$

(5)

The parameters for this kind of RVE were established according to the results of SEM images. However, we also performed a parametric study, considering a wide porosity spectrum from 25% to 95%.

Continuity conditions for the vector **w** in terms of quantity and flux (Eq (2)) have been implemented at the interface between the medium and the polymeric material. Moreover, a pointwise constraint has been fixed. The periodicity of the unit cell has been described through other continuity conditions between the opposite faces of the medium domain along the three spatial axes. In Fig 1B and 1D the main conditions for the microbeads scaffold in the *bcc* configuration (valid also for the *fcc* configuration) and for the fibers scaffold are reported.

The diffusivity of oxygen and glucose in the medium employed in this analysis are $3.83 \times 10^{-9}$ m$^2$/s and $9.59 \times 10^{-10}$ m$^2$/s respectively [18,46]. In the work of Zhao *et al*. the oxygen diffusion coefficient of alginate and chitosan microbeads has been estimated to be between $2.1 \pm 0.3 \times 10^{-9}$ m$^2$/s and $0.17 \pm 0.01 \times 10^{-9}$ m$^2$/s [47]. In this study, an intermediate value of $1.3 \times 10^{-9}$ m$^2$/s has been chosen. Regarding the diffusion of glucose through a chitosan matrix, the work of Singh and Ray was considered, which reported a diffusion coefficient ranging between $1 \times 10^{-10}$ and $1 \times 10^{-11}$ m$^2$/s [48] Thus, an intermediate value of $0.5 \times 10^{-10}$ m$^2$/s was selected for the actual model. While the diffusion coefficient of glucose in the PLA was not found in literature, an oxygen diffusivity of $7.6 \times 10^{-12}$ m$^2$/s has been identified for our study [49].

## Computational model of transport-consumption of nutrients and cell proliferation in a static culture

**Mathematical formulation of the model.** The evolution of cell cultures seeded within the previously described scaffolds was studied through the development of a continuous mathematical model of oxygen or glucose transport-consumption

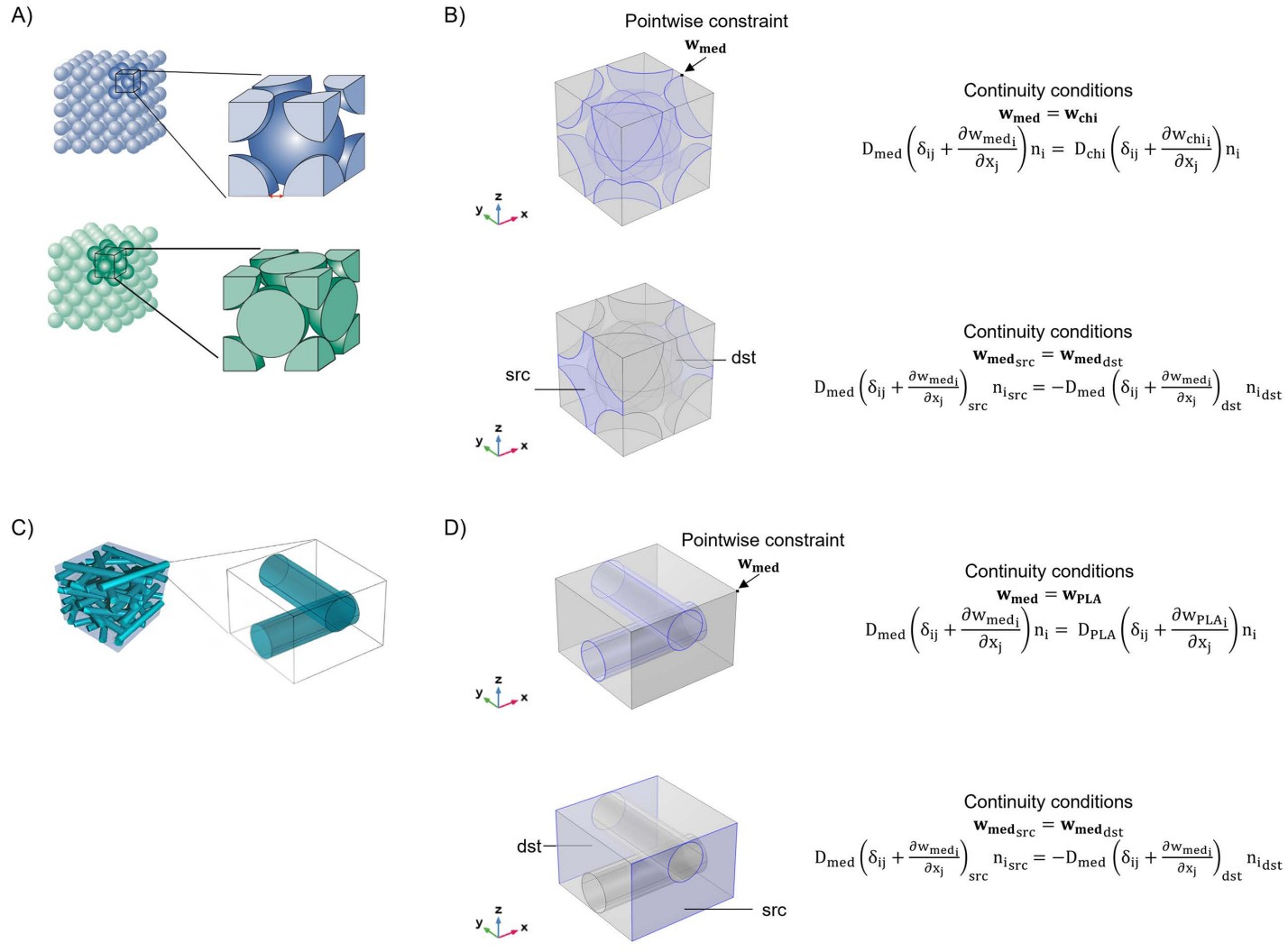

**Fig 1. Microarchitecture of the scaffolds employed for the simulations and boundary conditions for the corresponding RVEs. (A)** Spheres packing in body-centred cubic (*bcc*) configuration with a porosity of 32% and face-centred cubic (*fcc*) configuration with a porosity of 26% **(B)** Continuity equations at the interface between the culture medium and the chitosan microbeads (*bcc*); periodic condition of continuity between the source and the destination boundaries along the x-axis for beads **(C)** Randomic-oriented fibers and unit cell composed by perpendicular fibers **(D)** continuity equations at the interface between the culture medium and the PLA fibers; periodic condition of continuity between the source and the destination boundaries along the y-axis for PLA fibers; i,j = 1,2,3.

and cell proliferation through two different types of scaffolds seeded with astrocytes and neurons. The scaffold was put in a PDMS constraint to maintain the shape of scaffold components: in one case chitosan microbeads and in the other sheets of PLA fibers. The cell-laden scaffold was inserted in the incubator at T = 37°C, with a constant oxygen level of 0.2 mol/m³. The culture time was 21 days, during which the culture medium was changed every 3 days.

The PDMS ring is permeable to gas, such as oxygen [50], but not to large molecules, such as glucose. For this reason, a classic Fick law was employed for oxygen in the PDMS. Differently, the equation used to model the transport-consumption of the two nutrients in the scaffold domain is the following:

$$\frac{\partial s_i}{\partial t} = D_{s_i}\nabla^2 s_i - \alpha_{ia}s_i n_a - \alpha_{in}s_i n_n \tag{6}$$

where $s_i$ is the oxygen ($i = ox$) or glucose ($i = gl$) concentration, $D_{si}$ the diffusion coefficient of the nutrient through the homogenized scaffold, $\alpha_{ia}s_i$ and $\alpha_{in}s_i$ represent the of nutrient consumption rate for the single astrocyte and neuron, and $n_a$ and $n_n$ are the respective cell densities.

For a generic rat neuron, the maximum oxygen and glucose consumption rate are $7.54 \times 10^{-10}$ ml/min and $6.1 \times 10^{-9}$ μmol/min respectively [51], that correspond to $4.9 \times 10^{-16}$ mol/s, according to the ideal gas law at T = 37°C, and about $1 \times 10^{-16}$ mol/s. Regarding the rat astrocytes counterpart, we based the glucose consumption rate on Bouzier-Sore *et al.* [52], who found it to be about four times that of neurons. Based on the findings of Schuchmann *et al.* [53], who measured 5.7 nM/min for $10^7$ astrocytes in a 0.7 ml volume, the maximum oxygen consumption was calculated as $6.7 \times 10^{-17}$ mol/s. The different coefficients $\alpha_{ia}$ and $\alpha_{in}$ were calculated by normalizing the maximum nutrient consumption to the initial nutrient concentration.

The presence of cells could affect the nutrients diffusivity through the scaffold. Thus, it is possible to introduce an effective diffusion coefficient, according to the Maxwell's solution [54]

$$D_{eff,s_i} = D_{s_i} \frac{2(1-\theta)}{2+\theta}$$

(7)

where $\theta = (n_a + n_n)V_{cell}$ is the volume fraction occupied by cells that can be related to the cell size and density where $V_{cell}$ is the specific cell volume. For this first model neural cells were considered as spheres with a 10 μm diameter, neglecting the developing dendrites and the axons, in order to easily calculate the volume, starting from the cell body size [55]. Consequently, substituting the Eq. (7) in the Eq. (6) it has been possible to obtain the reaction/diffusion law for the two nutrients

$$\frac{\partial s_i}{\partial t} = D_{eff,s_i} \nabla^2 s_i - \alpha_{ia} s_i n_a - \alpha_{in} s_i n_n$$

(8)

The astrocyte density $n_a$ is represented by a combination of logistic cell growth and non-linear cell diffusion, enabling the inclusion of non-homogeneous seeding scenarios. In general, several nutrients are involved in the cell growth and the metabolic processes behind it are very complex especially when there is more than one phenotype. For simplicity, the growth has been made to depend only on oxygen.

$$\frac{\partial n_a}{\partial t} = D_{n_a} \nabla^2 n_a + \beta \left( s_{ox} - s_{ox,min} \right) n_a \left( 1 - \frac{n_a}{N_{max}} \right)$$

(9)

$$D_{n_a} (n_a) = D_{n_a}^* e^{\gamma (n_a - N_{max})}$$

(10)

$\beta(s_{ox} - s_{ox,min})$ is the cell proliferation rate, $s_{ox,min}$ is the hypoxic threshold identified for the cell culture [56]. Such formulation means that, while the levels of oxygen are higher than the threshold, cell growth is possible. $N_{max}$ is the maximum astrocyte density in the system. $D(n_a)$ [m²/s] represents non-linear cell diffusion, where $D_{na}^*$ is the cell diffusivity. In regions where $n_a \ll N_{max}$, cells do not spread appreciably, while from regions where the density is about equal to the maximum density ($n_a \leq N_{max}$) cells spread to lower density regions. $\gamma$ is a sensitivity parameter [29].

The model has been implemented in COMSOL Multiphysics. The geometry designed in an axisymmetric configuration is composed of the scaffold domain having a diameter of 5 mm and a PDMS ring with a width of 1.5 mm (Fig 2). The established height of the PDMS is 650 μm, based on the experimental set up, while the height of the scaffold depends on its modular components (beads or fibrous film). The equations have been written in their weak form in cylindrical coordinates using the *Weak Form PDE* interface. A triangular extra-fine mesh was generated.

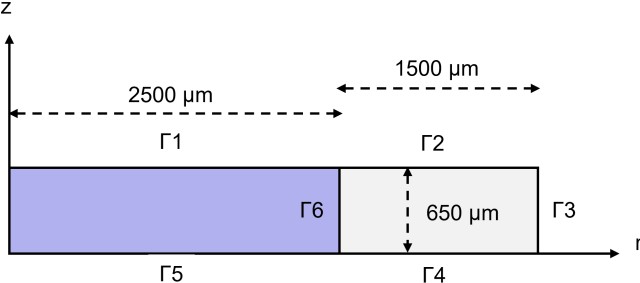

**Fig 2. 2D axisymmetric geometry of the microbeads scaffold within the PDMS constraint.** Sketch of the axisymmetric geometry of model composed of two domains: the PDMS (width: 1.5 mm) and the scaffold (radius: 2.5 mm). The two domains are 650 μm. The external and internal boundaries – Γ – are highlighted.

**Boundary and initial conditions.** The oxygen concentration on the top of the scaffold (Γ1, Γ2) and on the lateral surface (Γ3) is equal to the oxygen concentration in fresh culture medium (0.2 mol/m³) and it is kept constant by the incubator. The initial concentration is 0.2 mol/m³ everywhere. A zero-flux condition was imposed at the bottom (Γ4, Γ5).

$$-n \cdot \nabla s_{ox} = 0 \tag{11}$$

To simulate glucose delivery within a known volume of medium with a specific initial concentration, an ordinary differential equation (ODE) was introduced as a boundary condition on the top of the scaffold (Γ1). This ODE represents the flow of nutrients from the medium $V_m$ to the scaffold, through the exchange interface with area $A$ between the medium and the scaffold, as a function of cellular consumption.

$$\frac{ds_{gl,m}}{dt} V_m = \int D_{eff,s_{gl}} \frac{\partial s_{gl,sc}}{\partial t} dA \tag{12}$$

The initial glucose condition in the neural cell culture medium, supplemented with other factors, is 25 mol/m³. Furthermore, a medium refreshment has been simulated, resetting the glucose concentration every three days. From the literature, it is important to maintain this level above a critical threshold that, for glial and neural cells, was established to be 5 mol/m³ [57]. A zero-flux condition was set at the bottom (Γ5) and at the interface between the scaffold and the PDMS ring (Γ6).

$$-n \cdot \nabla s_{gl} = 0 \tag{13}$$

In a first study the initial cell density has been determined based on the number of cells seeded in the scaffolds. In the study conducted by Tedesco *et al.* [45] after the seeding the percentage of astrocytes was 5%; after 21 days the total cell density was unchanged but the percentage of neurons, due to cell death, had dropped to about 70% while the percentage of neurons of astrocytes, proliferating, increased to 30%. To account for apoptosis of neurons occurring in the first days of culture, a constant neuron density $N_n$ has been hypothesized. $N_{a0}$ and $N_{max}$ were derived respectively calculating 5% and 30% of the total cells cultured in the scaffold. No cell migration out of the scaffold has been hypothesized, thus a zero-flux condition was set at Γ1, Γ5 and Γ6.

$$-n \cdot \nabla n = 0 \tag{14}$$

For the 3D cell culture based on the chitosan microbeads, a first study has been conducted considering $3 \times 10^4$ microbeads and $1.2 \times 10^5$ total cells. Then, keeping constant the number of the microbeads, a parametric study was performed to evaluate the variations in oxygen levels as a function of the number of seeded cells, while maintaining a constant height. In this way it was possible to obtain the nutrient concentration within the construct varying the cell density.

As regards the PLA fibrous scaffold a first study has been conducted considering a single 60 μm high PLA sheet inserted in the PDMS constraint, considering $3 \times 10^4$ seeded cells. The considerations for the initial and maximum cell density have been supposed to be the same of the microbeads-based culture. The maximum number of PLA layers to limit the hypoxic region was calculated, keeping constant the initial cell number. Then the maximum cell number that can be seeded in the scaffold, maintaining constant the height of the construct but varying its porosity.

## Results

### *In vitro* 3D scaffold-based models for neuronal culture

The chitosan microbeads were previously characterized in the study of Tedesco *et al.* [45]. An IR spectroscopy analysis was conducted to verify the chitosan adhesion on the PLA film. The spectrum of the covered fibers was compared with the pure single materials. Interestingly, the spectrum of chitosan-coated PLA film obtained through filtration deposition (Fig 3Ac) shows the typical peaks of PLA (Fig 3Aa) and a deformation near 1640 cm$^{-1}$, in correspondence of the C=O double bond found in chitosan spectrum (Fig 3Ab), demonstrating the successful deposition of chitosan on the fibers.

The SEM analysis performed on the PLA films showed a random deposition of the fibers (Fig 3B). The mean diameter estimated with ImageJ was 2 μm and the mean porosity calculated was 74%.

The polymeric film seeded with glial and neuronal cells were fixed at the beginning and at the end of the culture. The cell-laden films were labelled with DAPI to confirm proper cell seeding at DIV 1 (Fig 3C). After 21 days of culture, they were treated with MAP2, a cytoskeletal marker, showed in green, and VGLUT, showed in red, to label synaptic vesicles, demonstrating a good cell adhesion and the formation of a dense neural network (Fig 3D).

### Homogenization theory

Considering $3 \times 10^4$ microbeads mixed with the cell suspension poured into the PDMS ring, occupying a volume $V_{tot} = 12.76$ mm$^3$, radii of 41 μm and 42 μm were determined for the *bcc* and *fcc* configurations, respectively, consistent with the experimental observations (Fig 1B). From the COMSOL simulation, two isotropic tensors, represented by a $3 \times 3$ diagonal matrixes with equal values along the diagonal, are obtained for the diffusivity in the microbeads-based unit cell, reflecting the cubic symmetry of the microstructure. Thus, single isotropic diffusion coefficients at the macro-scale are derived. The homogenized diffusivities obtained solving the Eq (1) are $1.71 \times 10^{-9}$ m$^2$/s and $2.47 \times 10^{-10}$ m$^2$/s for oxygen and glucose respectively in the *bcc* configuration $1.49 \times 10^{-9}$ m$^2$/s and $1.94 \times 10^{-10}$ m$^2$/s in the *fcc* one. These dense arrangements lead to the diffusivity within the homogenized scaffold assuming values closer to that of chitosan. Differently, due to the asymmetry of the RVE of the fibers scaffold, the diffusivity along the *x* and *y* directions is the same, while a different value was obtained for the *z* direction. Specifically, the homogenized oxygen diffusion coefficients are $2.38 \times 10^{-9}$ m$^2$/s on the *xy*-plane and $2.45 \times 10^{-9}$ m$^2$/s along the *z*-axis. In this case the values of the efficient diffusivity are closer to the diffusion coefficients in the medium, given the low diffusion of oxygen in the PLA fibers. For the glucose the values are $5.94 \times 10^{-10}$ m$^2$/s and $6.11 \times 10^{-10}$ m$^2$/s (Fig 4A). Moreover, the parametric study allowed us to evaluate the diffusivity through the RVE depending on the scaffold porosity, in turn dependent on the random fibers deposition. Specifically, we found a quite linear trend for this parameter (Fig 4B). It is interesting to observe a range between about 40% and 85% where the diffusivity along the *z*-axis is slightly higher than in the *xy*-plane. The same trend was found for the diffusion coefficient of glucose.

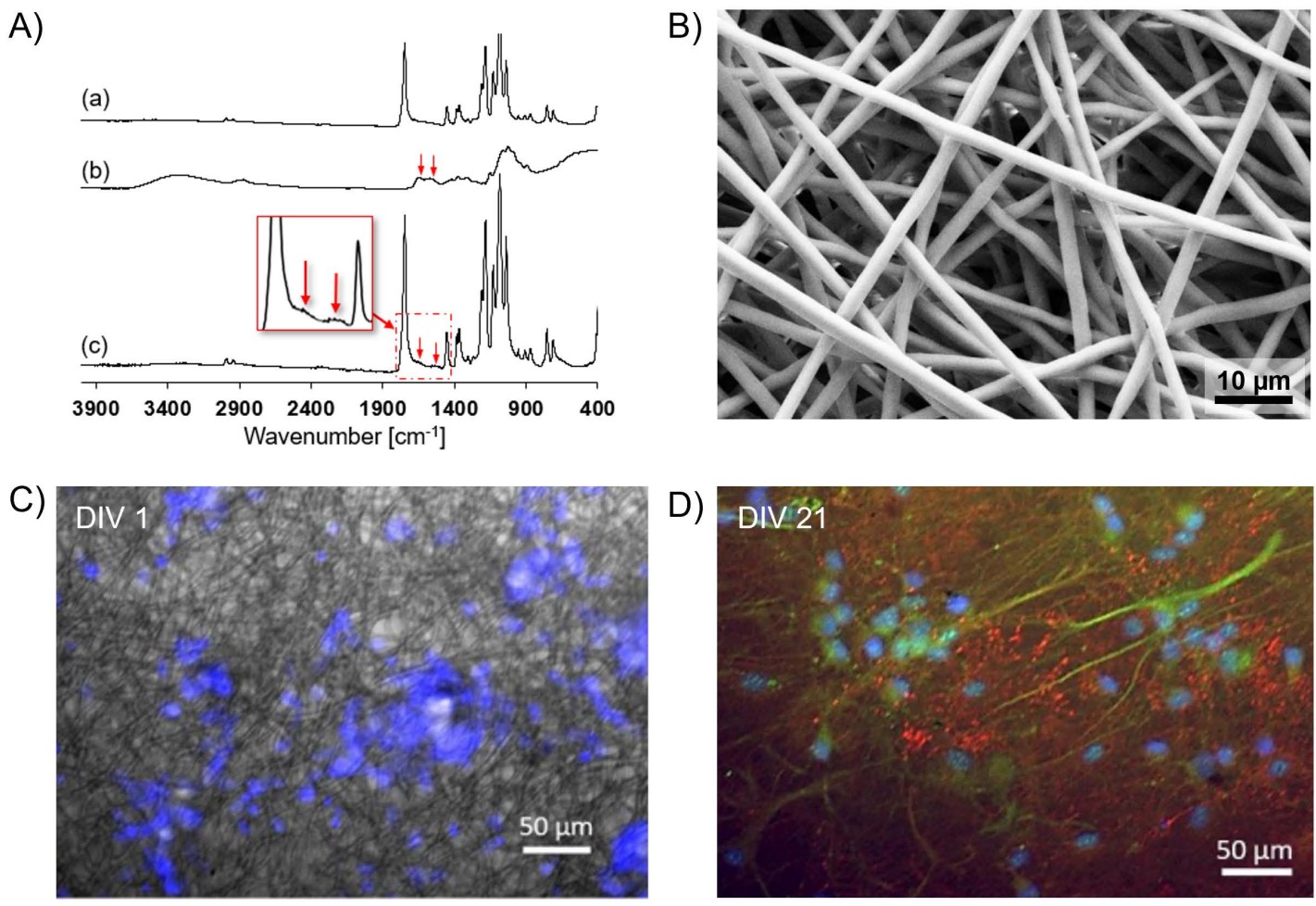

**Fig 3. Characterization of chitosan-covered PLA fibers. (A)** FTIR spectra of a) PLA; b) chitosan; c) electrospun PLA film treated with the chitosan via filtration deposition method **(B)** SEM image of a PLA film; scale bar 10 μm. **(C)** Nuclei of cells adhered to the substrate at DIV1; scale bar: 50 μm **(D)** Neuronal cytoskeleton marked with MAP2 (green) and glutamatergic synaptic vesicles marked with VGLUT (red) at DIV21; scale bar: 50 μm.

## Mathematical model of transport-consumption of nutrients and cell proliferation in a static culture

The proposed model predicts the nutrient gradient and the cell growth within 3D scaffolds based on chitosan microbeads assembly and PLA fibrous film. For all the 3D culture configurations a change of medium was simulated every 3 days. In the first studies the parameters regarding the cell density and the porosity of the scaffold have been taken from our experiments. Then it has been chosen to vary the number of cells seeded in the scaffold or the porosity of the model itself in the case of the PLA fibers-based construct. Cells located near the outer surface of the scaffold are exposed to a constant concentration of oxygen ($0.2\,mol/m^3$), as they are in direct contact with the surrounding environment, where oxygen given by the incubator is more readily available. Moving deeper into the scaffold, oxygen diffusion becomes limited. This leads to the formation of an oxygen gradient within the construct. It is possible to estimate the diffusion time on the bottom of the scaffold, in its inner part, as follows:

$$t_{diff} = \frac{h^2}{D_{s_{ox}}}$$

(15)

A)

| Diffusivity (m²/s) | | | | | | |
|---|---|---|---|---|---|---|
| | Medium | Microbeads | | | Fibers | | |
| | | Homogenized scaffold (bcc) | Homogenized scaffold (fcc) | chitosan | Homogenized scaffold (xy) | Homogenized scaffold (z) | PLA |
| oxygen | $3.83\times10^{-9}$ | $1.71\times10^{-9}$ | $1.49\times10^{-9}$ | $1.39\times10^{-9}$ | $2.38\times10^{-9}$ | $2.45\times10^{-9}$ | $7.6\times10^{-12}$ |
| glucose | $9.59\times10^{-10}$ | $2.47\times10^{-10}$ | $1.94\times10^{-10}$ | $0.5\times10^{-10}$ | $5.94\times10^{-10}$ | $6.11\times10^{-10}$ | 0 |

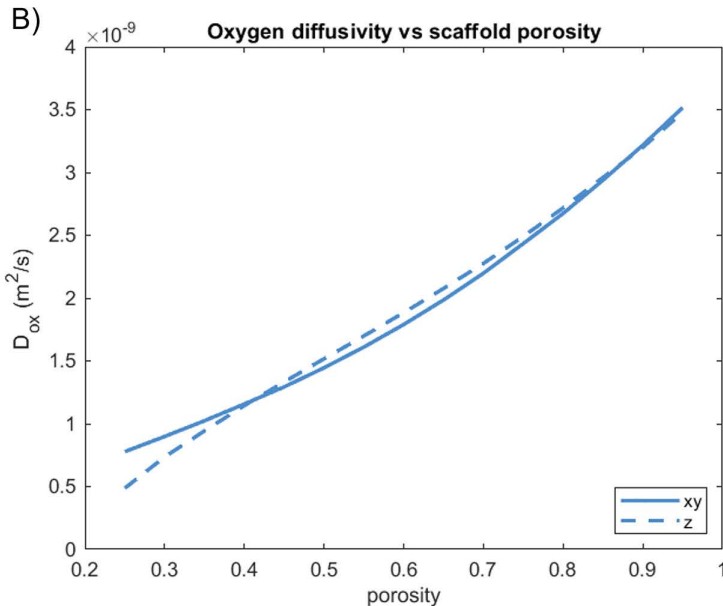

**Fig 4. Diffusivity of nutrients for the microbeads scaffold.** (A) Table with oxygen and glucose diffusivity in the culture medium, chitosan and homogenized scaffold in the two different microbeads packing and through the PLA fibers scaffold (B) Oxygen diffusivity as a function of porosity along xy (continuous line) and z direction (dashed line) in the fibers scaffold.

Initially, we decided to investigate the first hour of cell culture to predict the initial conditions shortly after seeding. Given the PDMS constraint, the scaffold's maximum height is 650 μm. Since the oxygen diffusivity is on the order of $10^{-9}$, diffusion in the deepest part of the construct occurs in less than 10 minutes. Our results show that, with constant oxygen tension in the incubator, a quasi-steady-state condition is established within 20 minutes. Fig 5 illustrates the oxygen gradients in the two different *in vitro* constructs and the corresponding nutrient profile along the symmetry axis.

Specifically, a 650 μm-high microbeads scaffold seeded with $1.2\times10^{5}$ neural cells ($9.4\times10^{12}$ cells/m³) maintains an oxygen level above the hypoxic threshold. The minimum oxygen concentrations reached in the scaffold core during this period were 0.0506 mol/m³ and 0.0439 mol/m³ for the *bcc* and *fcc* configurations (Fig 5A, 5B), with diffusion times of 4 and 5 minutes, respectively.

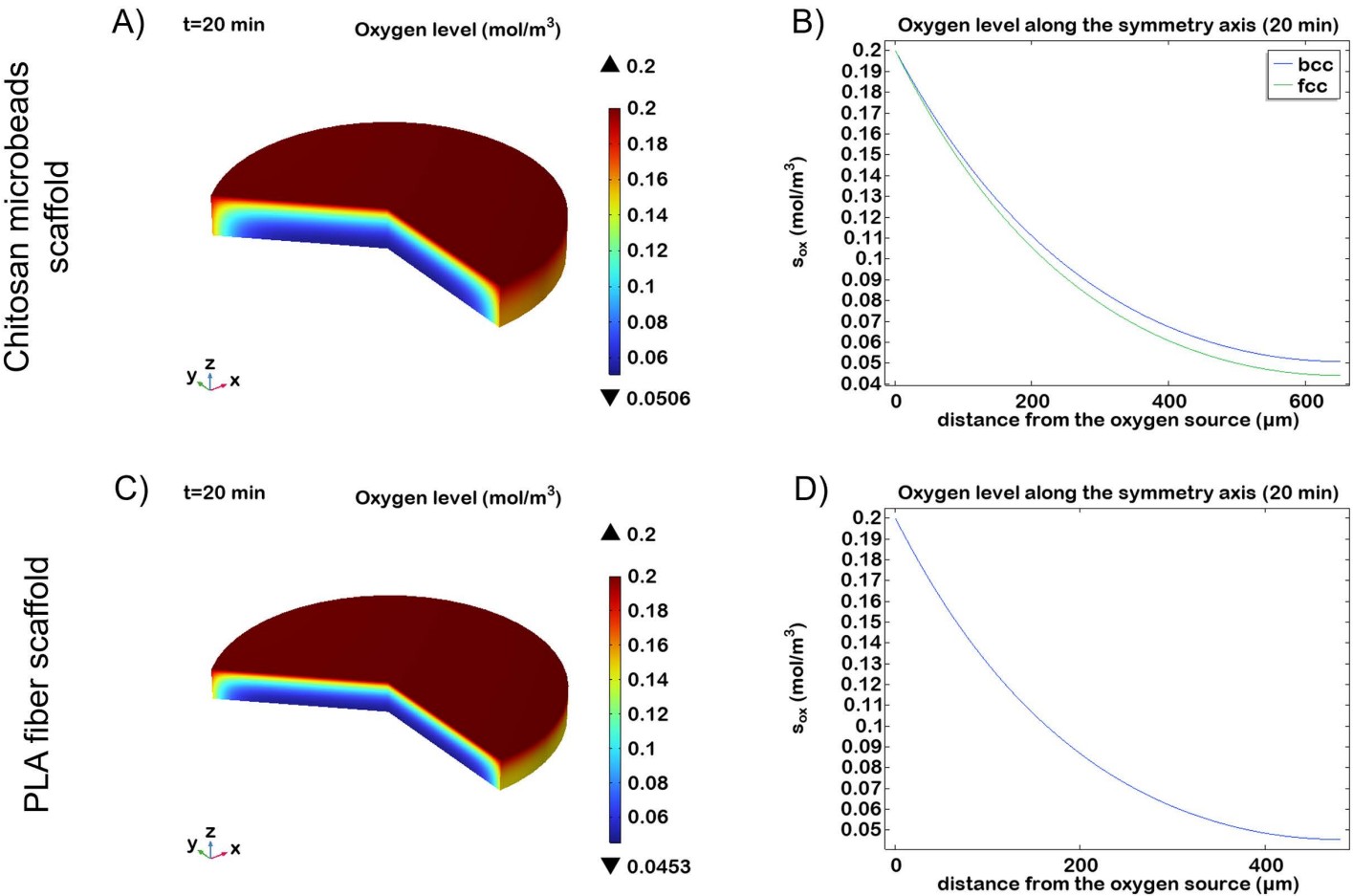

**Fig 5. Oxygen level within the microbeads scaffold and within the PLA fibers scaffold with a 74% porosity during the first 20 minutes of culture.** (A) 3D visualization of oxygen gradient in the microbeads scaffold with a *bcc* configuration (B) Oxygen concentration along the symmetry axis (r = 0) for the *bcc* (blue line) and *fcc* (green line) configuration (C) 3D visualization of oxygen gradient in the PLA scaffold (D) Oxygen concentration along the symmetry axis (r = 0) of the PLA scaffold.

Conversely, for neural cultures grown on a single 60 µm-thick PLA fiber film, seeded with $3 \times 10^4$ cells ($2.55 \times 10^{13}$ cells/m³), we estimated the maximum number of overlapping layers required to prevent hypoxia, reaching a total height of 480 µm. The calculated diffusion time was 1.5 minutes, with an oxygen concentration at the scaffold bottom of 0.0453 mol/m³ (Fig 5C, 5D).

Although the PLA fiber scaffold had a higher porosity (74%) than the microbeads-based counterparts (32% and 26%), its thickness is minor due to the higher predetermined cell density. These results suggest that, while diffusion is faster in thinner geometries, the higher cell density increase the risk of local hypoxia and may compromise cell health in deeper layers.

Together, these findings highlight the importance of scaffold architecture in tissue engineering applications. Considering these specific cell density, the microbead-based 3D scaffolds offer superior oxygenation profiles, enabling the construction of thicker neural tissues without compromising viability. In contrast, the fibrous film may require additional engineering strategies, to support cell function in multilayer configurations.

Following the observation of a well-oxygenated microenvironment conducive to cell growth immediately after seeding, the consumption of oxygen and glucose levels was systematically analysed throughout the entire cell culture period. In general, cells exhibited a higher oxygen consumption compared to glucose uptake. This is due to the predominant oxidative metabolism of neuronal cells, which are present in greater numbers than the initial glial cells, which, in contrast, display a more glycolytic metabolism. Microbeads scaffold, 650 µm in height and seeded with $1.2 \times 10^5$ cells ($9.4 \times 10^{12}$ cells/m$^3$), maintains an overall normoxic condition after 21 days of culture. The average oxygen concentration reaches approximately 0.107 mol/m$^3$ in the *bcc* arrangement and 0.102 mol/m$^3$ in the *fcc* arrangement, ensuring optimal conditions for both glial and neural cells. This indicates that oxygen diffusion is not significantly different between the two scaffold configurations. In both of case, at the end of cell culture the oxygen levels remain above the hypoxic threshold (Fig 6A).

Furthermore, the results show that during the entire culture the glucose concentration remains above the imposed threshold of 5 mol/m$^3$ owing not only to the low number of glycolytic cells but also to periodic medium refreshments (Fig 6B). Astrocytes exhibit a sigmoidal trend, according to the experimental data [45], with a lag phase up to day 5 and saturating the scaffold around day 15 (Fig 6C).

Regarding the PLA fiber scaffold, it is possible to observe values very close to those of the *fcc* configuration of the microbeads scaffold. At the end of the culture the minimum level achieved is about 0.041 mol/m$^3$ (Fig 6D), realizing a suitable condition for the astrocytes and the neurons with an average volumetric value of 0.097 mol/m$^3$. The glucose profile for the PLA fiber scaffold reaches lower concentrations due to the higher cell density seeded in the construct. The glucose seems to reach the imposed threshold of 5 mol/m$^3$ showing the importance of the medium refreshment (Fig 6E). Without this adjustment, neural cells would experience a hypoglycaemic condition, that would affect their functions.

Furthermore, it has been calculated the maximum height for the scaffold keeping constant the initial cell number. In particular, assuming a *bcc* packing configuration, the model predicts that a 750 µm high construct can be fabricated while maintaining adequate oxygenation, requiring approximately $3.5 \times 10^4$ microbeads (as estimated by Eq. (3)).

Fig 7A illustrates the oxygen concentration at the scaffold core as a function of the number of seeded cells, for both *bcc* and *fcc* arrangements. As expected, increasing the cell density leads to a progressive decrease in oxygen levels due to cellular consumption. Notably, when $1.4 \times 10^5$ cells are seeded in a *bcc* scaffold, oxygen levels in the core approach the critical hypoxic threshold of 0.04 mol/m$^3$ (horizontal red dashed line), with only a small hypoxic region forming (Fig 7B). However, a further increase to $1.6 \times 10^5$ cells causes a significant expansion of the hypoxic zone, with the oxygen concentration dropping to 0.036 mol/m$^3$ (Fig 7C).

The same conditions were then simulated for the *fcc* arrangement, to investigate the influence of scaffold geometry on oxygen transport. Interestingly, with $1.6 \times 10^5$ cells, the *fcc* configuration exhibits an oxygen profile (Fig 7D) very similar to that observed in the bcc model seeded with only $1.4 \times 10^5$ cells, both in terms of oxygen minimum and spatial extent of the hypoxic zone. This indicates that the *fcc* architecture, being denser, limits oxygen diffusion more severely, causing earlier onset of hypoxia at lower metabolic demand.

Finally, as shown in Fig 7E, when the *fcc* scaffold is seeded with $1.6 \times 10^5$ cells, the hypoxic region becomes more pronounced and widespread, with oxygen levels dropping as low as 0.0285 mol/m$^3$, well below the viability threshold. These results highlight how both the number of seeded cells and the internal scaffold geometry critically affect oxygen distribution, and suggest that *bcc* arrangements may offer improved conditions for cell viability under higher cell densities.

The parametric study conducted for the PLA-based *in vitro* model suggests that for a 480 µm scaffold, composed of stacked 60 µm-thick layers with a 74% of porosity, the maximum number of cells that can be seeded is about $3.1 \times 10^4$ per layer to guarantee normoxic conditions (Fig 8A). However, increasing the seeding density beyond this value leads to a significant drop in oxygen concentration, eventually falling below the viability limit and resulting in the presence of hypoxic regions. At a seeding density of $3.2 \times 10^4$ cells, the oxygen level in the core begins to approach the critical threshold, with small hypoxic zones emerging (Fig 8C). When the number of cells is increased further to $3.4 \times 10^4$, the hypoxic region becomes more extended, with oxygen concentrations dropping further below the threshold (Fig 8D). These findings

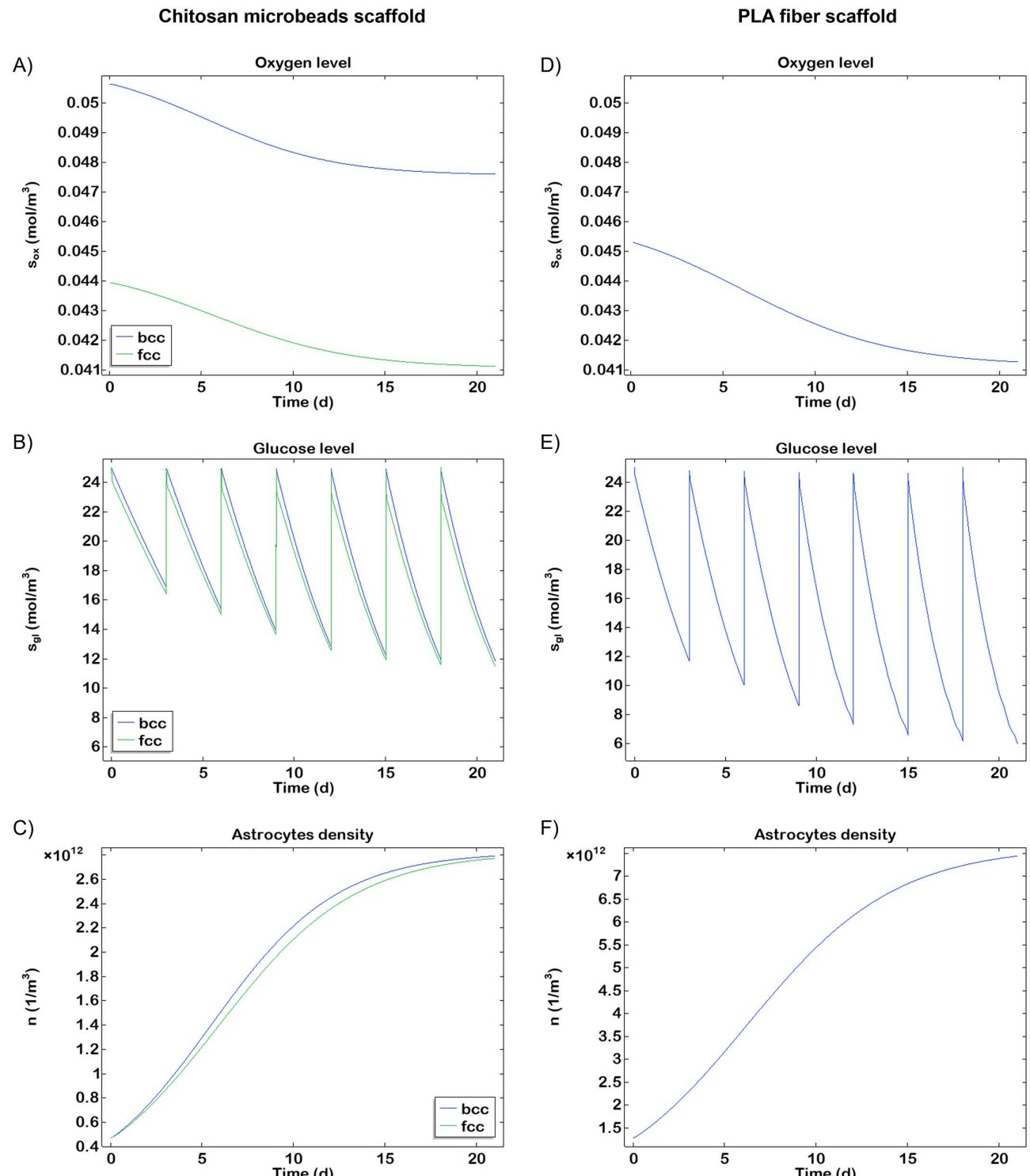

**Fig 6. Time evolution of a neural cell culture and associated variables in the chitosan microbeads scaffolds in the two different configurations (blue line for *bcc* and green line for *fcc*) and PLA fibers scaffold core with a 74% porosity.** (A) Oxygen level (B) Glucose level (C) Glial cell (astrocytes) density in the microbeads scaffold. The blue line represents the *bcc* configuration, and the green line represents the *fcc* configuration (D) Oxygen level (E) Glucose level (F) Glial cell (astrocytes) density in the PLA scaffold.

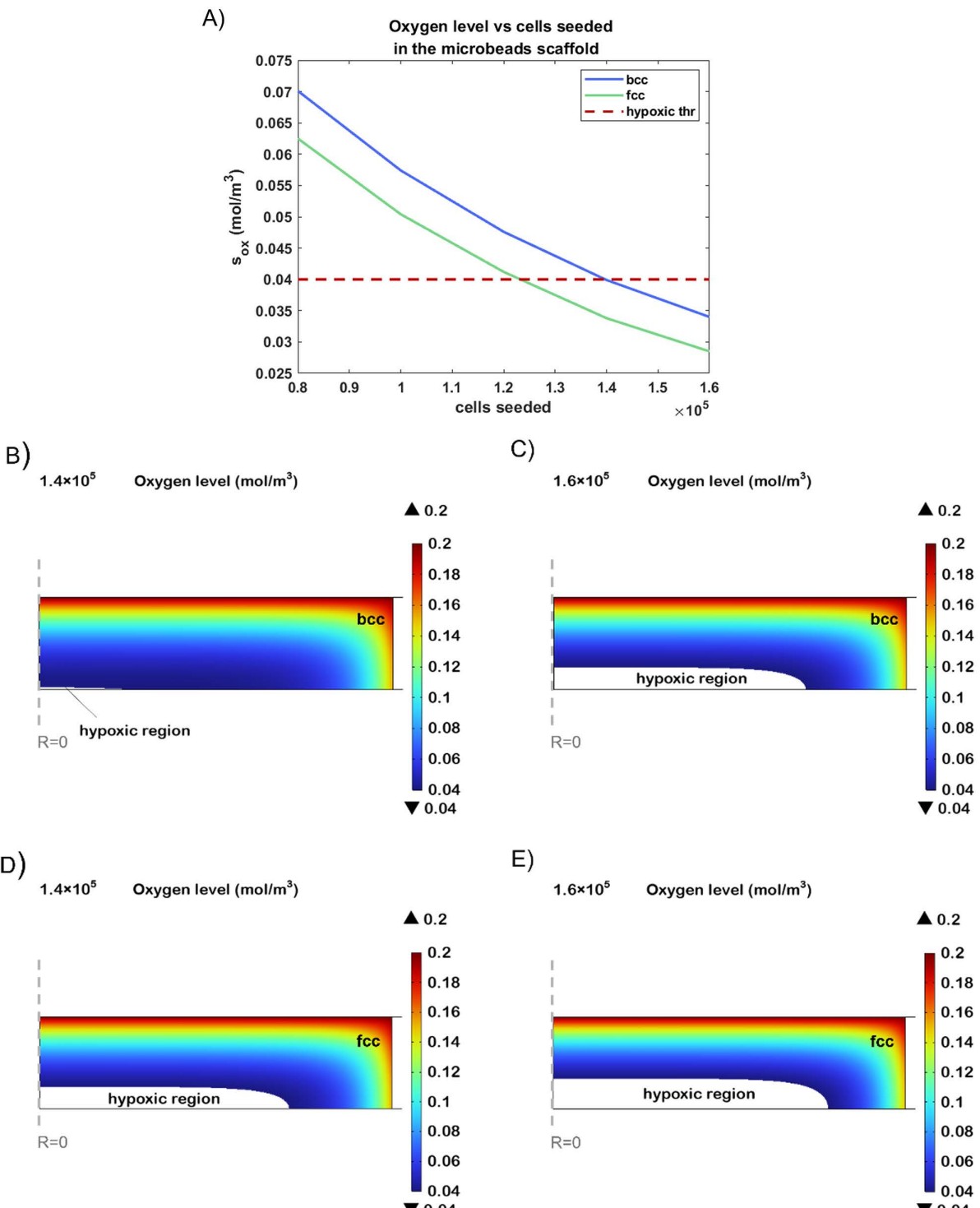

**Fig 7. Analysis of the scaffold varying the number of seeded cells in the microbeads scaffold.** (A) Oxygen concentration in the microbeads scaffold core in *bcc* (blue line) and *fcc* (green line) configuration in function of the number of cells seeded within the construct after 21 days of culture. Cross-sectional view of the scaffold seeded with $1.4 \times 10^5$ and $1.6 \times 10^5$ neural cells in *bcc* (B, C) and *fcc* (D, E) configuration after 21 days of culture.

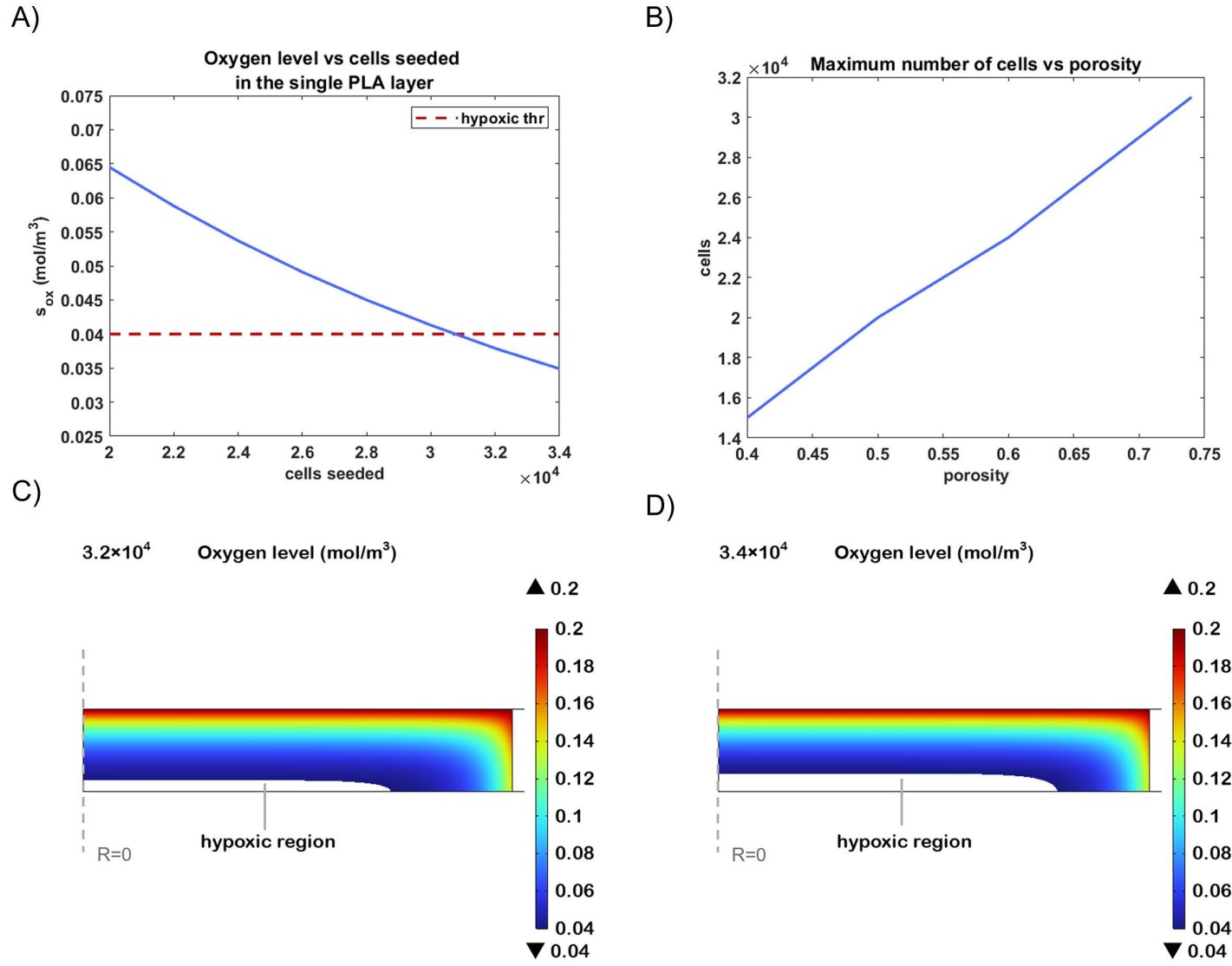

**Fig 8. Analysis of the scaffold varying the number of seeded cells in the microbeads scaffold varying the number of seeded cells in the PLA fibers scaffold.** (A) Oxygen concentration in the core in function of the number of cells seeded in the single 60 μm thick PLA layer, considering a 74% porosity. (B) Maximum number of cells which can be seeded to avoid hypoxia in function of the scaffold porosity. Cross-sectional view of the scaffold where each layer is seeded with $3.2 \times 10^4$ (C) and $3.4 \times 10^4$ total neural cells (D) after 21 days of culture.

indicate that even small increases in cell number beyond the optimal limit can critically compromise the oxygen availability. In addition, we evaluated how scaffold porosity influences oxygen transport capacity. Fig 8B shows the maximum number of cells that can be seeded while preserving normoxic conditions as a function of porosity. As expected, a higher porosity improves diffusivity and nutrient transport, allowing for a greater number of viable cells. In contrast, scaffolds characterized by a porosity of 40% can sustain only about $1.5 \times 10^4$ cells, due to limited interstitial space for oxygen diffusion.

These results highlight the critical interplay between scaffold architecture, porosity, and cellular metabolic demand. Specifically, they demonstrate how optimizing porosity is essential to ensure sufficient oxygenation and prevent hypoxia, particularly when designing scaffolds for high-density cell cultures or long-term in vitro applications.

## Discussion

The integration of computational tools facilitates a highly detailed analysis of scaffold porosity, material properties, and boundary conditions—factors that significantly influence both the nutrient profile and cellular behaviour. These analyses are indispensable for optimizing scaffold design, ensuring the delivery of adequate nutrients, and promoting uniform cell growth. Moreover, computational modelling reduces the need for labour-intensive and time-consuming experimental trials, by allowing the prediction of outcomes under various design scenarios. This not only accelerates the development process but also conserves valuable time and resources. The use of such modelling techniques not only advances the field of scaffold engineering but also contributes to the broader goal of developing reliable, efficient, and scalable tissue engineering solutions.

In the context of this study, a computational model was implemented to simulate nutrient diffusion and consumption, alongside the growth of neural cells. This model was applied to two distinct scaffolds, each representative of 3D *in vitro* nervous cultures, providing insights into how the design of scaffolds affects cellular behaviour and nutrient dynamics. Given the structural complexity of porous scaffolds, homogenization theory was used to simplify the analysis. From a computational point of view simulating the detailed microstructure of a porous scaffold at a direct level demands high spatial resolution and the solution of numerous equations, significantly increasing computational costs. This approach provides an equivalent macroscopic description of transport phenomena, avoiding the high computational cost of resolving microscale geometries directly. This approach, which was implemented in COMSOL Multiphysics, enabled the computation of homogenized macroscopic diffusion tensors for nutrient transport across the scaffolds. These tensors varied depending on the porosity of the periodic Representative Volume Element (RVE) used in the simulations.

While the methodology was applied to PLA fiber and chitosan microbead scaffolds, it is broadly applicable to any scaffold with a well-defined microscale structure. For simplification, microbeads were modelled as rigid spheres, neglecting their intrinsic elasticity. Thus, it became possible to propose an ordered arrangement and apply such mathematical procedure.

To the best of our knowledge, the application of homogenization theory is quite limited in modelling tissue-engineered constructs. In particular, only few studies focus on the simplification of complex microstructures in tissue engineering. For instance, although some papers have applied homogenization to calculate mechanical properties of scaffolds, studies addressing diffusive phenomena are rarer. Our work highlights the potential of homogenization theory in modelling nutrient diffusion within tissue engineered heterogeneous scaffold.

Initially, we conducted simulations based on the internal microstructure of the scaffolds, using experimental data regarding scaffold heights and neural cell densities to calculate the nutrient levels within the 3D constructs. These simulations also incorporated the dynamic aspect of the culture medium, which was refreshed every three days. Special attention was given to oxygen concentration levels, as maintaining adequate oxygen supply is crucial for preventing hypoxia and ensuring the physiological relevance of the culture conditions.

Subsequently, we analysed several variations in the scaffold design, including changes in the number of seeded neural cells, the porosity of the scaffolds, and the overall height of the structures. It is important to note that in the current computational model, the coupled effects of oxygen and glucose metabolism, essential in the metabolic processes of neurons and astrocytes, were not considered, as these interactions are inherently complex. Additionally, the ratio between neural and glial cells was kept constant throughout the simulations. Future studies will seek to address these metabolic complexities by incorporating the effects of oxygen-glucose interactions. This will enable a more comprehensive representation of *in vivo*-like behaviour, particularly from a metabolic perspective, and help to enhance the reliability of 3D *in vitro* models for neuroengineering applications.

In conclusion, the study presents a novel strategy for designing bottom-up scaffolds for tissue engineering, employing the mathematical framework of homogenization. The biofabrication parameters and methods established in this work offer significant potential for optimizing the growth and functionality of the glial-neural 3D network, laying a solid foundation for

future developments in neuroengineering and tissue regeneration. Furthermore, by addressing the lack of simplified modelling approaches for microscopically complex structures in tissue engineering, our study paves the way for more efficient and scalable tissue engineering solutions.

## Author contributions

**Conceptualization:** Laura Pastorino, Donatella Di Lisa.

**Data curation:** Ilaria Parodi.

**Formal analysis:** Ilaria Parodi.

**Methodology:** Ilaria Parodi, Giacomo Damonte, Donatella Di Lisa.

**Supervision:** Laura Pastorino, Silvia Scaglione, Marco Massimo Fato, Donatella Di Lisa.

**Validation:** Donatella Di Lisa.

**Visualization:** Ilaria Parodi, Donatella Di Lisa.

**Writing – original draft:** Ilaria Parodi.

**Writing – review & editing:** Laura Pastorino, Giacomo Damonte, Silvia Scaglione, Donatella Di Lisa.

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
