## [Decision Letter · Decision Letter 0]

9 Jun 2025

*in vitro*

PLOS ONE

Dear Dr. Parodi,

Thank you for submitting your manuscript to PLOS ONE. After careful consideration, we feel that it has merit but does not fully meet PLOS ONE’s publication criteria as it currently stands. Therefore, we invite you to submit a revised version of the manuscript that addresses the points raised by the reviewers during the review process.

We look forward to receiving your revised manuscript.

Kind regards,

Pradeep Kumar, Ph.D.

Academic Editor

PLOS ONE

2. Please include a caption for figures Fig 2, Figure. 3Ac, Fig 8A.

3. Please ensure that you refer to Figure 4 in your text as, if accepted, production will need this reference to link the reader to the figure.

Additional Editor Comments (if provided):

Reviewers' comments:

Reviewer's Responses to Questions

**Comments to the Author**

1. Is the manuscript technically sound, and do the data support the conclusions?

Reviewer #1: Yes

Reviewer #2: Yes

2. Has the statistical analysis been performed appropriately and rigorously?

Reviewer #1: N/A

Reviewer #2: Yes

3. Have the authors made all data underlying the findings in their manuscript fully available?

Reviewer #1: Yes

Reviewer #2: Yes

4. Is the manuscript presented in an intelligible fashion and written in standard English?

Reviewer #1: Yes

Reviewer #2: Yes

Reviewer #1: This manuscript presents a computational framework for analyzing nutrient transport and cell proliferation within 3D porous scaffolds designed for in vitro neural cultures. By applying homogenization theory and reaction-diffusion modeling in COMSOL, the authors estimate effective diffusivity and simulate the spatiotemporal distribution of oxygen, glucose, and cell density across different scaffold architectures and porosities. The study is technically robust and offers potential interest for PLOS ONE readers. However, while the work is promising and methodologically detailed, several issues must be addressed before it can be considered for publication.

• The integration of homogenization theory with nutrient transport modeling is a valuable approach. However, the manuscript lacks a clear articulation of its novelty in comparison to existing literature. The authors should explicitly state the methodological advancements and explain how this framework contributes to scaffold modeling in the context of neural tissue engineering.

• Several paragraphs, particularly in the Introduction and Discussion, are dense and contain overlapping or tangential information. Reorganizing the content will help highlight the study's core contributions and findings. For instance, the Introduction paragraph beginning with “Another type of 3D culture is represented by porous scaffolds...” jumps between mechanical stiffness, hypoxia, and metabolic waste without clear transitions. Streamlining these sections will improve readability and focus.

• Comparisons with experimental data, such as oxygen levels or cell viability, are mostly qualitative. Where possible, the authors should provide quantitative benchmarks or error analyses to support model validation. For example, the statement “The calculated diffusion time was 1.5 minutes...” is informative, but its relevance should be discussed in the context of experimental oxygen measurements or biological outcomes such as viability or proliferation.

• Key modeling assumptions such as treating neural cells as spheres and neglecting dendritic or axonal extensions are briefly mentioned but not critically evaluated. For example, in the Methods section: “...neural cells were considered as spheres with a 10 μm diameter, neglecting the developing dendrites and the axons...” The potential impact of this simplification on transport dynamics and cell behavior predictions should be discussed.

• Several figures (e.g., Figs 5, 6, and 7) contain rich and informative data but are not adequately explained in the Results section. Each figure should be contextualized with a clear description of what it illustrates and its implications. For instance, Fig 5D presents oxygen concentration along the scaffold's symmetry axis, but the biological relevance—such as thresholds for proliferation or hypoxia-induced apoptosis—is not discussed.

• Some sentences are overly long or contain multiple ideas, which hinders comprehension. Additionally, there are minor grammatical and phrasing issues throughout the manuscript. For instance, the sentence “...was chosen as unit cell” should be revised to “was chosen as the unit cell.”

Reviewer #2: This article "Computational analysis of 3D biopolymeric porous scaffolds for the in vitro development of neural networks" requires major revisions, including language editing for clarity and coherence. Additionally, a more comprehensive literature review is necessary to contextualize the findings and strengthen the overall argument.

How do the biosorption capabilities of Aspergillus niger polysaccharides compare to those of traditional water remediation materials?

What methodologies were employed to elucidate the structure of the polysaccharides, and how do these methods enhance our understanding of their functionality?

Can you elaborate on the tailoring of the polysaccharides' functionality for specific contaminants in water remediation?

What are the potential environmental implications of using Aspergillus niger spore polysaccharides in water treatment processes?

What challenges or limitations are associated with the large-scale application of these biosorbents in real-world water remediation scenarios?

-More physical explanation of results is required.

-The Figures quality are too weak please improve the quality and put some arrays on the important part

What specific biopolymeric materials were used in the scaffolds, and what are their advantages for neural tissue engineering?

How does the porosity of the scaffolds affect nutrient diffusion and cellular behavior within the neural networks?

What comparisons were made between the computational analysis results and experimental findings, and how do they validate the study's conclusions?

Are there any limitations in the computational analysis that could affect the scalability or applicability of the scaffolds in clinical settings?

---

## [Decision Letter · Decision Letter 1]

12 Oct 2025

Computational analysis of 3D biopolymeric porous scaffolds for the *in vitro* development of neural networks

PONE-D-25-20544R1

Dear Dr. Parodi,

We’re pleased to inform you that your manuscript has been judged scientifically suitable for publication and will be formally accepted for publication once it meets all outstanding technical requirements.

Kind regards,

Pradeep Kumar, Ph.D.

Academic Editor

PLOS ONE

Additional Editor Comments (optional):

Reviewers' comments:

Reviewer's Responses to Questions

**Comments to the Author**

Reviewer #1: All comments have been addressed

Reviewer #2: All comments have been addressed

2. Is the manuscript technically sound, and do the data support the conclusions?

Reviewer #1: (No Response)

Reviewer #2: Yes

3. Has the statistical analysis been performed appropriately and rigorously?

Reviewer #1: (No Response)

Reviewer #2: Yes

4. Have the authors made all data underlying the findings in their manuscript fully available?

Reviewer #1: (No Response)

Reviewer #2: Yes

5. Is the manuscript presented in an intelligible fashion and written in standard English?

Reviewer #1: (No Response)

Reviewer #2: Yes

Reviewer #1: (No Response)

Reviewer #2: (No Response)

**Do you want your identity to be public for this peer review?** For information about this choice, including consent withdrawal, please see our Privacy Policy

Reviewer #1: No

Reviewer #2: No

---

## [Editor Report · Acceptance letter]

PONE-D-25-20544R1

PLOS ONE

Dear Dr. Parodi,

I'm pleased to inform you that your manuscript has been deemed suitable for publication in PLOS ONE. Congratulations! Your manuscript is now being handed over to our production team.

Kind regards,

on behalf of

Prof. Pradeep Kumar

Academic Editor

PLOS ONE